# Characterization of Living Dental Pulp Cells in Direct Contact with Mineral Trioxide Aggregate

**DOI:** 10.3390/cells9102336

**Published:** 2020-10-21

**Authors:** Tamaki Hattori-Sanuki, Takeo Karakida, Risako Chiba-Ohkuma, Yasuo Miake, Ryuji Yamamoto, Yasuo Yamakoshi, Noriyasu Hosoya

**Affiliations:** 1Department of Endodontology, School of Dental Medicine, Tsurumi University, 2-1-3 Tsurumi, Tsurumi-ku, Yokohama 230-8501, Japan; 2911001@stu.tsurumi-u.ac.jp (T.H.-S.); hosoya-n@tsurumi-u.ac.jp (N.H.); 2Department of Biochemistry and Molecular Biology, School of Dental Medicine, Tsurumi University, 2-1-3 Tsurumi, Tsurumi-ku, Yokohama 230-8501, Japan; karakida-t@tsurumi-u.ac.jp (T.K.); chiba-r@tsurumi-u.ac.jp (R.C.-O.); yamamoto-rj@tsurumi-u.ac.jp (R.Y.); 3Department of Anatomy, School of Dental Medicine, Tsurumi University, 2-1-3 Tsurumi, Tsurumi-ku, Yokohama 230-8501, Japan; miake-y@tsurumi-u.ac.jp

**Keywords:** mineral trioxide aggregate, dental pulp cells, fluorescent labeling, cell chemotaxis, calcium phosphate crystal

## Abstract

Mineral trioxide aggregate (MTA) was introduced as a material for dental endodontic regenerative therapy. Here, we show the dynamics of living dental pulp cells in direct contact with an MTA disk. A red fluorescence protein (DsRed) was introduced into immortalized porcine dental pulp cells (PPU7) and cloned. DsRed-PPU7 cells were cultured on the MTA disk and cell proliferation, chemotaxis, the effects of growth factors and the gene expression of cells were investigated at the biological, histomorphological and genetic cell levels. Mineralized precipitates formed in the DsRed-PPU7 cells were characterized with crystal structural analysis. DsRed-PPU7 cells proliferated in the central part of the MTA disk until Day 6 and displayed a tendency to move to the outer circumference. Both transforming growth factor beta and bone morphogenetic protein promoted the proliferation and movement of DsRed-PPU7 cells and also enhanced the expression levels of odontoblastic gene differentiation markers. Mineralized precipitates formed in DsRed-PPU7 were composed of calcium and phosphate but its crystals were different in each position. Our investigation showed that DsRed-PPU7 cells in direct contact with the MTA disk could differentiate into odontoblasts by controlling cell–cell and cell–substrate interactions depending on cell adhesion and the surrounding environment of the MTA.

## 1. Introduction

Mineral trioxide aggregate (MTA) is a material for dental endodontic therapy developed in the United States in the early 1990s and it was marketed as a ProRoot^®^ MTA (Dentsply Tulsa Dental, Tulsa, OK, USA). In Japan, MTA was approved as a pulp-capping material and has been on the market since 2007. This material is hydraulic modified Portland cement that was originally developed as root-end filling material. In addition to these applications, MTA has been applied to direct pulp capping, vital pulpotomy, root canal fillings and root resorption repair [1,2]. Therefore, it has attracted attention as a new biomaterial for pulp capping instead of calcium hydroxide, which has been used until now.

MTA is mainly composed of inorganic oxides such as tricalcium silicate, dicalcium silicate and tricalcium aluminate containing bismuth oxide and plaster [3,4,5,6]. During the hydration of MTA, calcium hydroxide and calcium silicate hydrate are produced and released from MTA. Studies related to these components were conducted on the biological functionality of MTA, such as biocompatibility [1,2,3,7,8], sealing ability [1,2], antibacterial qualities [9,10], tissue engineering [11] and the mechanism of inducing apatite formation [12,13,14,15,16,17]. In addition to research related to the characteristics of MTA using mesenchymal stem cells and human dental pulp cells, many studies were conducted to investigate the effect of MTA on cytotoxicity [18,19], cell proliferation [20,21,22], gene expression [21,23,24] and apoptosis [25]. However, there are few reports that examined the dynamics of living cells in direct contact with the MTA. 

The immortalized porcine dental pulp cell line (PPU7) was established [26] and has been used for endodontic research associated with laser irradiation [27]. PPU7 cells can express alkaline phosphatase (ALP) and odontoblastic marker genes that form precipitated nodules when cultured in a medium containing β-glycerophosphate and ascorbic acid. Transforming growth factor beta (TGF-β) is a signaling molecule that upregulates odontoblastic marker genes; dentin sialophosphoprotein and matrix metalloprotease 20 in PPU7 cells suggest that this cell can differentiate into odontoblasts [28]. Here, we report the direction to dentin formation by analyzing cell proliferation, chemotaxis, crystalline status, the effect of growth factors and gene expression with visualized living PPU7 cells in direct contact with an MTA disk.

## 2. Materials and Methods

The present study was approved by the Institutional Ethics Review Committee of the Tsurumi University School of Dental Medicine (Yokohama, Japan), Recombination DNA Experiment and Biosafety Committee of the Tsurumi University School of Dental Medicine (project identification code 1318, 1 December 2015). All experiments were conducted based on relevant guidelines and regulations.

### 2.1. Gene Transfection of Discosoma Species Red Fluorescent Protein (DsRed) into a Porcine Dental-Pulp Cell Line (PPU7)

The PPU7 cell line was established from porcine dental pulp cells by our group [26]. The pDsRed–Express–DR vector (Takara Bio USA Inc., Mountain View, CA, USA) containing a cytomegalovirus promoter in a multicloning site was used for gene transfection. Cells were plated at subconfluent cell densities and transfected with a pDsRed–Express–DR vector using Lipofectamine 2000 in accordance with the manufacturer’s instructions (Thermo Fisher Scientific, Waltham, MA, USA). Two days after transfection, cells were replated at low density and cultured until colonies were visible. Individual colonies consisting of stable fluorescent (DsRed-PPU7) cells were isolated with cloning cylinders and maintained in alpha Minimum Essential Medium (αMEM; Gibco/Life Technologies, Carlsbad, CA, USA) containing 10% fetal bovine serum (FBS) and antibiotics (50 U/mL of penicillin, 50 μg/mL of streptomycin, Gibco; standard medium) at 37 °C in a humidified 5% CO_2_ atmosphere. Then, DsRed-PPU7 cells were cultured in standard medium at 37 °C in a humidified 5% CO_2_ atmosphere. PPU7 cells without DsRed labeling were used as the control and were cultured in αMEM containing 10% fetal bovine serum (FBS) and the antibiotics described above.

### 2.2. Alkaline Phosphatase (ALP) Activity Assay of the PPU7 Cell Line

Both PPU7 and DsRed-PPU7 cells were plated on a 96-well plate at a density of 1.0 × 10^4^ cells/well and cultured in standard medium for 24 h. The medium was then changed to a fresh standard medium. Measurement of ALP activity in each well was described previously [29]. After 72 additional hours of incubation, cells were washed once with PBS and ALP activity was assayed using 10 mM p-nitrophenylphosphate as the substrate in 100 mM 2-amino-2-methyl-1,3-propanediol-HCl buffer (pH 10.0) containing 5 mM MgCl_2_ and incubated for 4 minutes at room temperature. Addition of 0.2 M NaOH quenched the reaction and absorbance at 405 nm was read on a plate reader.

### 2.3. Changes in the Number of DsRed-PPU7 Cells on the Mineral Trioxide Aggregate (MTA) Disk

The MTA (ProRoot MTA, Dentsply Sirona Tulsa Dental Specialties, Johnson City, TN, USA) was mixed with sterile water according to the manufacturer’s instructions. An MTA disk with a 10 mm diameter and 1.8 mm thickness was prepared using a silicone mold and dried in a fume hood for 24 h followed by storage in a desiccator. Prior to the preparation of MTA disk, glass paste plates, spatulas and silicone molds for kneading were immersed in 100% ethanol and ultrasonically treated for 5 min in a fume food. The MTA disk was immersed in 5 mL of the standard medium at 37 °C in a humidified 5% CO_2_ atmosphere and DsRed-PPU7 (1.0 × 10^4^) cells were gently seeded on the surface of the MTA disk. Cells were cultured for 10 days at 37 °C in a humidified 5% CO_2_ atmosphere with the standard medium being changed every other day. On Days 1, 3, 5, 7, 8 and 10, the MTA disk was gently placed with the cell-adhesive side down in a culture medium that had been placed in advance in a 6-well plate that was thinly coated with 0.5% agarose gel and cells were observed in a living state under an inverted fluorescence microscope (Section A.1). The number of cells was counted by randomly selecting nine squares (300 μm × 300 μm) using Image J software Version 1.52a (National Institutes of Health, Bethesda, MD, USA).

### 2.4. Correlation of Fluorescence Intensity and Cell-Proliferation Rate of DsRed-PPU7 Cells

DsRed-PPU7 cells were plated on a 96-well plate at a density of 1250 to 20,000 or 2000 cells/well and cultured in the standard medium for 24 h. Then, the medium was changed to a fresh standard medium and cells were cultured for 6 days at 37 °C in a humidified 5% CO_2_ atmosphere; the standard medium was changed every other day. DsRed-PPU7 cells were photographed with an exposure time of 1/8 seconds using an inverted fluorescence microscope (Biozero BZ-8100, Keyence, Osaka, Japan) and fluorescence intensity was analyzed using Image J software. Subsequently, the proliferation rate against the same cells was determined with a CellTiter 96^®^AQuous One Solution Cell Proliferation Assay (MTS assay; Promega Corporation, Madison, WI, USA). The coefficient of determination (R^2^) was calculated from a regression line prepared on the basis of fluorescence intensity or the cell proliferation rate against cell density or cell-culture days.

### 2.5. Characterization of DsRed-PPU7 Cells on MTA Disks

The MTA disks were plated on a 6-well plate and immersed in 5 mL of the standard medium for 3 h at 37 °C in a humidified 5% CO_2_ atmosphere. After changing to the new standard medium, a glass cloning cylinder with an internal diameter of 6 mm was placed in the center of the disk and 2.0 × 10^4^ DsRed-PPU7 cells were gently seeded on the surface of the MTA disk in the cylinder. After the cells were left on the surface for 1 h, the cylinder was gently removed and cells were cultured at 37 °C in a humidified 5% CO_2_ atmosphere. On the next day, the medium was changed to a growth medium containing 10 mM β-glycerophosphate and 50 μM ascorbic acid in standard medium supplemented with 500 ng/mL of recombinant human bone morphogenetic protein 2 (rhBMP-2) [30], 50 nM LDN-193189 (Tocris Bioscience, Bristol, UK) which is a selective BMP signaling inhibitor, 1 ng/mL of human recombinant transforming growth factor β (rhTGF-β) or 1 μM SB-431542 (ChemScene, Monmouth Junction, NJ, USA) which is a selective inhibitor of TGF-β Type I receptor activin receptor-like kinase A. The medium was changed every two days and cells were cultured for 14 days at 37 °C in a humidified 5% CO_2_ atmosphere. DsRed-PPU7 cells were observed with the inverted fluorescence microscope as described in Section 2.3. Cells were photographed every 2 days for 14 days with an exposure time of 1/20 s.

### 2.6. Quantitative Polymerase Chain Reaction (qPCR) Analysis

Sample preparation for qPCR analysis was carried out based on our previously method [31]. Cells were extracted with a High Pure RNA Isolation Kit (Roche Diagnostics GmbH, Mannheim, Germany) and the purified total RNA (0.5 μg) was reverse-transcribed. Then, the reaction mixture composed of SYBR Green PCR Master Mix (Roche Diagnostics GmbH) was supplemented with 0.5 μM forward and reverse primers and 0.67 μL of cDNA as a template. For the specific primer setting, a Primer-BLAST (https://www.ncbi.nlm.nih.gov/tools/primer-blast/index.cgi) was used as a primer-designing tool. The specific primer sets and running conditions are shown in Table A1. Glyceraldehyde-3-phosphate dehydrogenase (*Gapdh*) was used as the reference gene. Each ratio was normalized to the relative quantification data of dentin sialophosphoprotein variant 1 (*Dspp-v1*) and 2 (*Dspp-v2*), matrix metalloprotease 20 (*Mmp20*) and Type I collagen alpha 1 chain (*Col1a1*) in comparison to *Gapdh*, which was generated on the basis of a mathematical model for relative quantification in the qPCR system.

### 2.7. Scanning Electron Microscopy (SEM) and Electron-Probe Microanalysis (EPMA)

On Day 14 of the cell culture, the MTA disk was cut with a razor to prepare the specimens for scanning electron microscopy (SEM) and electron-probe microanalysis (EPMA). The MTA disk was rinsed with PBS, fixed with 2.5% glutaraldehyde in 100 mM cacodylate buffer, dehydrated in ethanol and immersed in hexamethyldisilazane. Then, the surfaces were air-dried, mounted on aluminum stubs and sputter-coated with a gold layer of 300 Å thickness using an ion coater (Quick Auto Coater SC-701AT, Sanyu Electron Co., Ltd., Tokyo, Japan). Specimens were examined with a JCM-6000 (JEOL Ltd., Tokyo, Japan) scanning electron microscope.

The fixed MTA disk was substituted by propylene oxide and embedded in epoxy resin. Specimens were sliced to a thickness of approximately 1 mm using a diamond-coated rotating saw on a precision-cutting machine (Buehler IsoMet Low Speed Saw, Chicago, IL, USA). Specimens were mounted on aluminum stubs and sputter-coated with a carbon layer of 200 Å thickness using a carbon coater (Meiwafosis Co., Ltd., Tokyo, Japan). Elemental mapping of the mineralized precipitates was carried out using a JEM-8900R (JEOL Ltd., Tokyo, Japan) electron probe microanalyzer at an accelerating voltage of 20 kV and an irradiation current of 2.5 × 10^–8^ mA.

### 2.8. Statistical Analysis

All values are presented as the mean ± standard error of the mean (SEM). Statistical significance was determined using the Mann-Whitney U test for both ALP and MTS assay and Student’s t-test for the qPCR. In all cases, *p* < 0.05 was regarded as statistically significant.

## 3. Results

### 3.1. Comparison of PPU7 and DsRed-PPU7 Cells

In order to investigate the dynamics of dental pulp cells on an MTA disk, we used a porcine dental pulp-derived cell line (PPU7) for the present study (Section A.2) [26]. As a first step, we attempted to visualize PPU7 cells on MTA by introducing the DsRed–Express–DR vector into PPU7 cells. We isolated colonies consisting of stable fluorescent (DsRed-PPU7) cells after liposome transfection (Figure 1a). Subsequently, we were able to obtain stable fluorescent cells by cloning with the limiting-dilution method (Figure 1b). The inherent ALP activity of PPU7 cells with or without DsRed labeling was almost at the same level and there was no significant difference between PPU7 and DsRed-PPU7 cells (Figure 1c). We interpreted this finding as evidence that DsRed-PPU7 cells could be used in this study without problems.

### 3.2. Changes in the Number of DsRed-PPU7 Cells on the MTA Disk

We next prepared the MTA disk (Figure 2a) and cultured DsRed-PPU7 cells on it with a standard medium for 10 days. For cell observation, we gently placed the cell-adhesion surface of the MTA disk on a plate thinly coated with 0.5% agarose gel (Figure 2b) (Section A.1). We counted the number of cells up to Day 10 under fluorescence microscopy using Image J software (Figure 2c) and graphed changes in the number of DsRed-PPU7 cells over time (Figure 2d). The number of cells was approximately 200 cells/mm^2^ on Day 1 but it was decreased to approximately 40 cells/mm^2^ on Day 5. The number of cells gradually increased from Day 6 onward and it recovered to almost the same number as on Day 1 on Day 9. On Day 10, the number of cells reached 1.25 times that on Day 1. Here we faced the problem that an accurate cell number could not be obtained due to cells overlapping on the MTA disk with the increase in cells. In order to solve this problem, we attempted to measure the fluorescence intensity of the cells.

### 3.3. Correlation between Fluorescence Intensity and Cell-Proliferation Rate of DsRed-PPU7 Cells

We investigated whether there was a correlation between cell density and fluorescence intensity and between cell density and the cell proliferation rate. We seeded DsRed-PPU7 cells on a 96-well plate at a density of 1250 to 20,000 cells/well (Figure 3a) and analyzed fluorescence intensity using Image J software the next day (Figure 3b), followed by measurement of the cell proliferation rate for the same cells by MTS assay (Figure 3c). The coefficient of determination (R2), calculated from a regression line, was 0.997 between fluorescence intensity and cell density and 0.939 between the cell proliferation rate and cell density.

We next tested whether there was a correlation between cell-culture days and fluorescence intensity or the cell proliferation rate. We seeded DsRed-PPU7 cells on a 96-well plate at a low density of 2000 cells/well and cultured them for 6 days (Figure 4a) while measuring fluorescence intensity (Figure 4b) and the cell proliferation rate (Figure 4c) every day.

The fluorescence intensity and cell proliferation rate of DsRed-PPU7 cells on Day 5 increased by 21.8 and 18.8 times, respectively, as compared with those on Day 0 but they did not increase much on Day 6. Therefore, we calculated the coefficient of determination from a regression line up to Day 5 to be 0.972 between fluorescence intensity and cell-culture days (Figure 4d) and 0.9839 between the cell proliferation rate and cell-culture days (Figure 4e).

Since there was a positive correlation between the fluorescence intensity and cell proliferation rate of DsRed-PPU7 cells, we used the fluorescence intensity for the evaluation of the cell proliferation ability in the present study.

### 3.4. DsRed-PPU7 Cells on the MTA Disk

Because a significant decrease in the number of DsRed-PPU7 cells was detected on Day 5 when cells were seeded directly using pipettes on the MTA disk (Figure 2b), we improved the culture method to observe DsRed-PPU7 cells on the MTA disk over time as follows. We soaked the MTA disk in the standard medium for 3 h and seeded the DsRed-PPU7 cells in a fresh standard medium using a cloning cylinder (Figure 5a). On the next day, we changed the medium to a fresh growth medium, followed by a 14-day culture period. Based on fluorescence microscopy, the fluorescence intensity of the DsRed-PPU7 cells possessed a strong signal in the central part of the MTA disk until Day 6 but the intensity was strong on the outside of the MTA disk after Day 10 (Figure 5b,c). The fluorescence intensity of the central part of the MTA disk on Day 6 was 4.25 times higher than that on Day 0. The fluorescence intensity of the central part and the outside of the MTA disk was reversed after Day 11 and the outside intensity on Day 14 was 37.5 times higher than that on Day 0 (Figure 5d). With the decrease in fluorescence intensity of the central part of the MTA disk, mineralized precipitates were evident at that location on Day 14, although no mineralized precipitates were observed on the disk cultured without cells (Figure 5e).

### 3.5. Effect of BMP and TGF-β on DsRed-PPU7 Cells on the MTA Disk

To further investigate the dynamics of DsRed-PPU7 cells on the MTA disk, we cultured DsRed-PPU7 cells in the presence or absence of rhBMP-2 or rhTGF-β1. Based on fluorescence microscopy, the presence of rhBMP-2 and rhTGF-β1 enhanced the fluorescence intensity of DsRed-PPU7 cells in the central part of the MTA disk on Days 6–8 (Figure 6a). Figure 6b shows the change in fluorescence intensity over time in the central part of the MTA disk. When the fluorescence intensity of DsRed-PPU7 cells on Day 0 was set to 1.0, the fluorescence intensity in the presence of rhTGF-β1 on Day 6 was the highest (4.72-fold) and it was dramatically inhibited in the presence of SB-431542, which is a selective inhibitor of TGF-β Type I receptor activin receptor-like kinase A. The presence of rhBMP-2 also increased the fluorescence intensity (3.90-fold) on Day 8 but the fluorescence intensity was suppressed in the presence of LDN-193189, which is a selective BMP signaling inhibitor. After 10 days, the cells cultured in the presence of rhTGF-β1 or rhBMP-2 began to move to the outer circumference. Even cells cultured in only the growth medium (i.e., the control) displayed the same phenomenon. On Day 14, the fluorescence intensity of DsRed-PPU7 cells in the central part of the MTA disk cultured in the presence of rhTGF-β1 or rhBMP-2 was at almost the same level as that of the control. Interestingly, the fluorescence intensity of DsRed-PPU7 cells in the central part of the MTA disk cultured in the presence of LDN-193189 or SB-431542 was almost at the same level as that cultured in the standard medium on Day 14. Those fluorescence intensities were higher than those of the control, rhTGF-β1 or rhBMP-2.

### 3.6. Direction of Differentiation of DsRed-PPU7 Cells on the MTA Disk 

In order to observe the direction of cell differentiation, we next investigated gene expression in DsRed-PPU7 cells on Day 14. The gene expression of a panel of odontoblastic markers in DsRed-PPU7 cells on Day 14 was analyzed using qPCR (Figure 7). The expression levels of *Dspp-v1, Mmp20 and Col1a1* in cells cultured with rhTGF-β1 were significantly higher (2.41-fold for *Dspp-v1*, 3.62-fold for *Mmp20 and* 1.87-fold for *Col1a1*) than those of the control. These mRNA levels were dramatically inhibited in the presence of SB-431542. The gene expression of *DSPP-v2* was also slightly increased in the presence of TGF-β1 but there was no significant difference compared to the control. 

The expression level of *Dspp-v1, Mmp20 and Col1a1* in cells cultured with rhBMP-2 was also significantly higher (1.26-fold for *Dspp-v1*, 4.65-fold for *Mmp20 and* 1.46-fold for *Col1a1*) than that of the control and this level was suppressed in the presence of LDN-193189.

### 3.7. Observation of Mineralized Precipitates on the MTA Disk

We further attempted to characterize the mineralized precipitates formed on the MTA disk. Both stereoscopic microscopy and scanning electron microscopy (SEM) of the mineralized precipitates on MTA disk on Day 14 revealed the presence of spherical precipitates on the surface when cells were cultured in only the growth medium (control) and in the presence of LDN-193189, rhTGF-β1 and SB-431542 (Figure 8 and Figure A1). A number of spherical precipitates were formed when cells were cultured in LDN-193189. Interestingly, few mineralized precipitates were observed on the MTA disk cultured in the presence of rhBMP-2.

We selected cells cultured in LDN-193189 for further characterization (Figure 9 and Figure A2). The SEM of the mineralized precipitates on the MTA disk mainly showed spherical precipitates with an approximate 110 μm diameter on Day 14 (Figure 9a,b). In addition to those precipitates, triangular-pyramid-shaped precipitates were observed but their size was considerably smaller than that of the spherical precipitates (Figure 9a,c). No mineralized precipitates were observed on the MTA disk when the cells were cultured in the standard medium (Figure 9d). Moreover, a number of platelike crystals formed on the surfaces of both spherical and triangular-pyramid-shaped precipitates (Figure 9e,f).

Figure 10 shows electron-probe microanalysis (EPMA)-based elemental-mapping images of representative spherical and triangular-pyramid-shaped precipitates. The spherical precipitates possessed a cavity inside and no element was detected there. The mineral precipitates on the outer circumference were composed of Ca and P (Figure 10a). The triangular-pyramid-shaped precipitates, on the other hand, possessed a high concentration of Ca inside and their outer circumference comprised mineral precipitates consisting of Ca, P and spherical precipitates (Figure 10b). No Si was detected in the spherical and triangular-pyramid-shaped precipitates. The Ca/P molar ratios of each point, calculated on the basis of fluorite, were 0.82 for A, 1.15 for B, 1.95 for C, 1.63 for D, 1.69 for E and 2.01 for F.

## 4. Discussion

In research on MTA, approximately 60 papers have been reviewed by a comprehensive literature survey via electronic database of PubMed/MEDLINE [32]. In studies using MTA disk, the effects of cells on elution components from disks in the same environment as that of the cells and disks with a culture insert were investigated [19,23,33]. In addition, research was carried out to examine the effects of cells seeded on the plate on eluants extracted by immersing the disks in a culture medium [21]. However, there were few reports that directly show the dynamics of living cells in contact with MTA because the direct observation of cells is difficult, as the MTA is an impermeable cured body. Our challenge was to directly observe living PPU7 cells on an opaque MTA disk.

pDsRed–Express–DR is a eukaryotic expression vector that encodes a variant of the Discosoma sp. red fluorescent protein [34]. This protein was engineered for increased solubility and reduced cytotoxicity [35]. We demonstrated that the gene transfection of the DsRed protein into PPU7 cells did not affect cell morphology or the retention of inherent ALP activity (Section A.3). This meant that DsRed-PPU7 cells could be used without problems in subsequent experiments.

MTA can generally be used as dental repair material with little cytotoxic injury [1,2,3,8]. However, as one of the biological properties of the MTA, when a cured MTA body is soaked in water, the pH of the solution for the sustained release of calcium hydroxide is maintained at about 12 [5,6,36,37,38]. Because of the high surface pH of the MTA, MTA causes serious cell damage to soft tissue [39]. In fact, a study using stem cells from exfoliated deciduous teeth revealed that MTA impaired cell viability and caused cell apoptosis when cells were in direct contact with MTA [25]. In the present study using DsRed-PPU7 cells, we demonstrated that cell numbers were markedly reduced after Day 3. This finding suggests that inhibition of cell growth (i.e., apoptosis/necrosis) after stimulation with MTA could be caused by the release of calcium hydroxide from the material and the induced increase in pH.

Caspase consists of extrinsic, intrinsic or perforin/granzyme pathways that cause apoptosis in cells [40,41]. During the execution phase of cell apoptosis, the caspase-3 promotes compensatory cell growth through the production of prostaglandin E2 [42,43]. In fact, we demonstrated that DsRed-PPU7 cells proliferated after Day 5 and the number of cells reached the same number on Day 10 as that on the first day. This suggests that vital pulp therapy using MTA helps dental pulp cells adapt to an environment with harmful side effects, such as apoptosis.

In general, since adhesion cells are known to overlap between cells, DsRed-PPU7 cells on MTA disks should also be considered to always be overlapping. Therefore, there was a possibility that ambiguity would occur in the count of the number of cells with respect to cell proliferation. In order to avoid this risk, we examined whether the fluorescence intensity of cells could be replaced with the cell proliferation rate. We demonstrated that there was correlation between cell proliferation and fluorescence intensity. These results enabled us to evaluate the growth rate of cells in our subsequent experiments using fluorescence intensity.

Cell chemotaxis shows positive or negative cell motility depending on the concentration gradient of chemoattractants or chemorepellents. It was reported that dentin and pulp extracellular-matrix components containing a growth factor induce the chemotaxis of dental pulp cells [44,45]. A recent study showed that MTA promotes the chemotaxis of immune-cell migration through the activation of calcium-sensing receptors [46]. We demonstrated that cells proliferated in the central portion of the MTA disk until Day 6 after the seeding of the DsRed-PPU7 cells but then cell density increased outside the MTA disk on Day 14. Moreover, mineralized precipitates were notably observed in the central portion of the MTA disk on Day 14. These findings suggest that the mineralized precipitates were produced in the presence of DsRed-PPU7 cells and the negative chemotaxis of DsRed-PPU7 cells occurred due to environmental changes in association with mineralized precipitates.

TGF-β regulates dental pulp cell differentiation and odontoblastic differentiation, which are related to ectoderm–mesenchyme molecular interactions [47,48,49,50]. MTA contains tricalcium silicate and dicalcium silicate and the hydration (i.e., curing reaction) of those components produces calcium hydroxide and calcium silicate hydrate. Therefore, MTA and calcium hydroxide dissolve the mineral components of dentin and it is associated with releasing TGF-β remains embedded in the matrix [51]. Subsequently, TGF-β leads to activate its signaling transduction pathways to promote the formation of tertiary dentinogenesis [52]. In some studies, calcium–silicate-containing silicon dioxide was shown to stimulate the TGF-β/Smad pathway [53,54]. The present study demonstrated that TGF-β1 increased fluorescence intensity on Day 8 in the central part of the DsRed-PPU7 cells on the MTA disk in the presence of TGF-β1. In addition to this, we demonstrated that fluorescence intensity of the central part of the MTA disk cultured in the presence of rhTGF-β1 or rhBMP-2 was almost at the same level as that of the control on Day 14. This finding may suggest that neither rhTGF-β1 nor rhBMP-2 affects the movement speed of DsRed-PPU7 cells. Since our findings indicate that TGF-β1 certainly enhances the cell proliferation, it is necessary to investigate the ability of cell proliferation in the presence of MTA using dentin disk as future studies.

Two *Dspp* splice variants (*Dspp-v1* and *Dspp-v2)* (Section A.4) and matrix metalloproteases 2 and 20 (MMP2 and MMP20) are able to be candidates for odontoblast differentiation markers [28]. That study further revealed that TGF-β1 enhanced the gene expressions of *Dspp-v1, Dspp-v2 and Mmp20* [28]. The present study demonstrated that the mRNA levels of *Dspp-v1, Mmp20 and Col1a1* were increased in DsRed-PPU7 cells cultured in the presence of rhTGF-β1. This finding is exactly in agreement with the results of our previous research and suggests that DsRed-PPU7 cells may have differentiated in the direction of odontoblast-like cells. In addition, our finding suggests that BMP-2 enhances the expression of odontoblastic marker genes, that is, the differentiation of odontoblast-like cells.

In general, in cases where cultured cells cannot adhere onto the carrier, apoptosis signaling occurs in the cells. To avoid this, spheroids are formed by the assembly and aggregation of cells and they secure a scaffold for each other. Even if cells adhere onto the carrier, in the case that adhesion is weak, spheroids are formed by assembly and aggregation due to cell migration on the basis of the intracellular mechanical action of the cell–substrate interaction. Because spheroids formed by a single cell do not have a vascular network such as tissues and organs in vivo, a concentration gradient of material is formed inside the spheroid. As a result, the cell niche surrounding the cell drastically changes and necrosis occurs in the central part where oxygen cannot be supplied [55,56]. We demonstrated that the spherical precipitates had a cavity inside in which no elements were detected. This finding suggests that necrosis may have occurred due to the intracellular concentration distribution of substances in DsRed-PPU7 cells, cell–cell interactions and mechanical actions associated with spheroid formation. Moreover, we demonstrated that the crystals on the surface of spherical precipitates were composed of Ca and P but Ca/P molar ratios were 0.82 to 1.15. Based on these Ca/P ratios [57], our result suggests that the platelike crystals on the surface of spherical precipitates were possibly calcium phosphate crystals, such as dicalcium phosphate dihydrate, in the process of transition from the amorphous phase.

When immersing the MTA-cured body in a phosphate buffer, calcium phosphate crystals containing an apatite-like structure on the surface of MTA are precipitated by the reaction of released calcium ions from the MTA and phosphate ions in the buffer [8]. Moreover, the sustained release of Ca ions from the cured MTA body in aqueous solutions forms a Ca-leached layer on the MTA surface [58]. We observed a high Ca concentration layer inside the triangular-pyramid-shaped precipitates containing no P or Si and crystals composed of Ca and P on the Ca-leached layer. Considering the Ca/P ratios at each point (1.63–2.01) [57], our finding suggests that some of the crystals on the triangular-pyramid-shaped precipitates may possess an apatite-like structure.

## 5. Conclusions

One of the novelties of this study was that the observation of cells on the MTA was possible through the introduction of fluorescent protein genes. With this approach, we were able to prove that there was a correlation between cell density and fluorescence intensity and between cell density and the cell proliferation rate. Moreover, we demonstrated the cell proliferation, cytotoxicity and chemotaxis, as well as the effect of cytokines, crystals and direction of differentiation of DsRed-PPU7 cells. Thus, our investigation showed that DsRed-PPU7 cells in direct contact with the MTA disk could differentiate into odontoblasts by controlling cell–cell and cell–substrate interactions depending on cell adhesion and the surrounding environment of the MTA. Further studies and animal experiments using human dental pulp cells are required to clarify these aspects.

## Figures and Tables

**Figure 1 cells-09-02336-f001:**
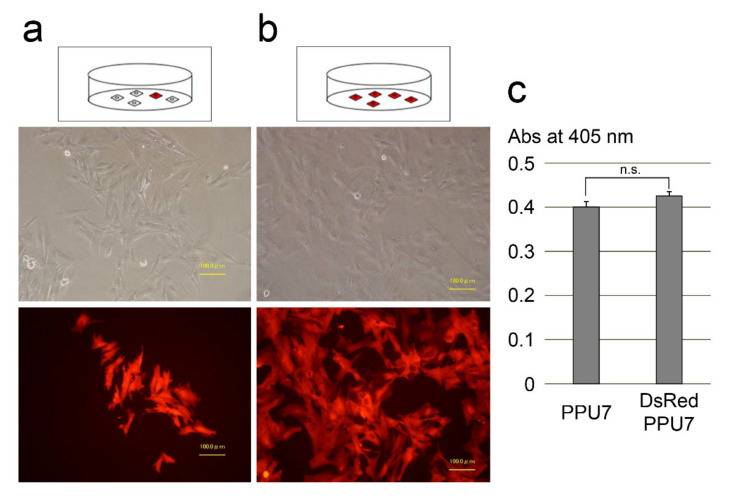
Gene transfection of Discosoma species’ red fluorescent protein (DsRed) into a porcine dental pulp cell line (PPU7). (**a**) DsRed-PPU7 cells immediately after liposome transfection (scale bar: 100 μm). (**b**) DsRed-PPU7 cells obtained by cloning with limited dilution (scale bar: 100 μm). (**c**) Inherent alkaline phosphatase (ALP) activity in cells before (PPU7) and after (DsRed-PPU7) gene transfection (*n* = 6). n.s., not significant difference.

**Figure 2 cells-09-02336-f002:**
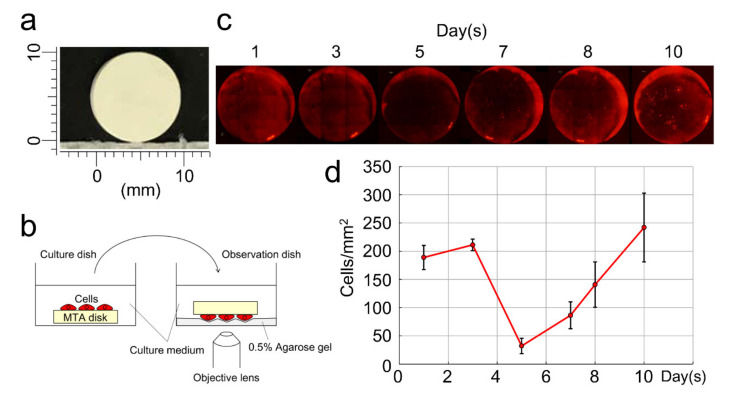
Cytotoxicity of mineral trioxide aggregate (MTA) based on the cell proliferation rate of DsRed-PPU7 cells cultured in standard medium. (**a**) MTA disk constructed using a silicon mold (10 mm in diameter with 1.8 mm thickness). (**b**) Schema for observation of living DsRed-PPU7 cells on MTA disks using agarose gel. (**c**) Fluorescence microscope image of DsRed-PPU7 cells on MTA disks on Days 1, 3, 5, 7, 8 and 10. (**d**) Changes in the number of DsRed-PPU7 cells on Days 1, 3, 5, 7, 8 and 10. Cells were counted by randomly selecting nine squares (300 μm × 300 μm) using Image J (*n* = 6).

**Figure 3 cells-09-02336-f003:**
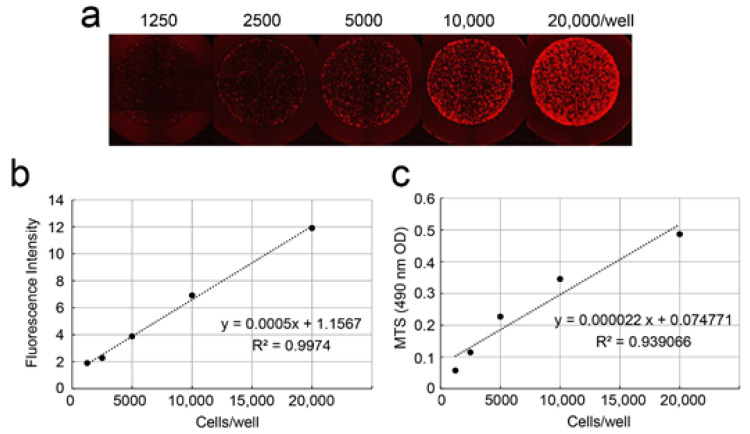
Correlation of fluorescence intensity and the cell proliferation rate with the cell density of DsRed-PPU7 cells. (**a**) Fluorescence microscope image of DsRed-PPU7 cells seeded on a 96-well plate at a density of 1250, 2500, 5000, 10,000 and 20,000 cells/well. (**b**) Regression line showing the correlation of the fluorescence intensity with cell density. (**c**) Regression line showing the correlation of cell proliferation with cell density by MTS assay.

**Figure 4 cells-09-02336-f004:**
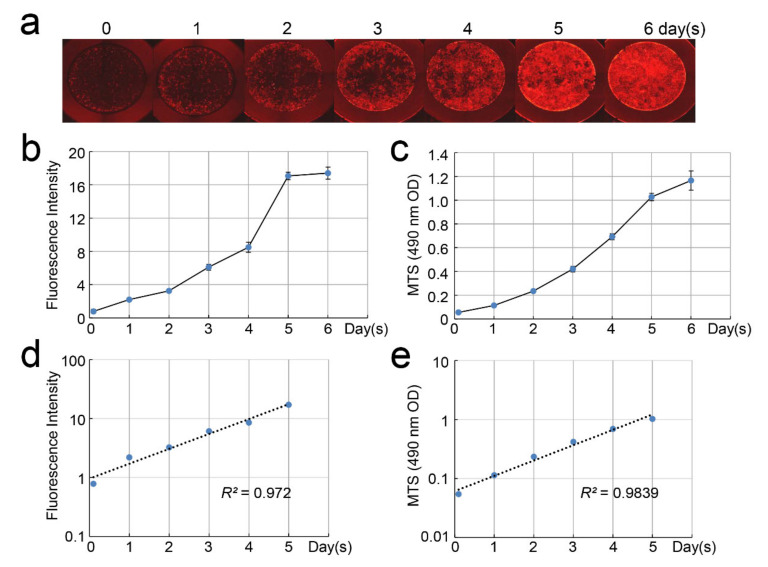
Correlation of fluorescence intensity and the cell proliferation rate with number of cell-culture days. (**a**) Fluorescence microscope image of DsRed-PPU7 cells seeded on a 96-well plate at 2000 cells/well and cultured for 6 days. (**b**) Changes in the fluorescence intensity of DsRed-PPU7 cells (*n* = 6). (**c**) Changes in the cell proliferation rate of DsRed-PPU7 cells based on MTS assay (*n* = 6). (**d**) Regression line showing the correlation of fluorescence intensity with culture days. (**e**) Regression line showing the correlation of the cell proliferation rate with culture days.

**Figure 5 cells-09-02336-f005:**
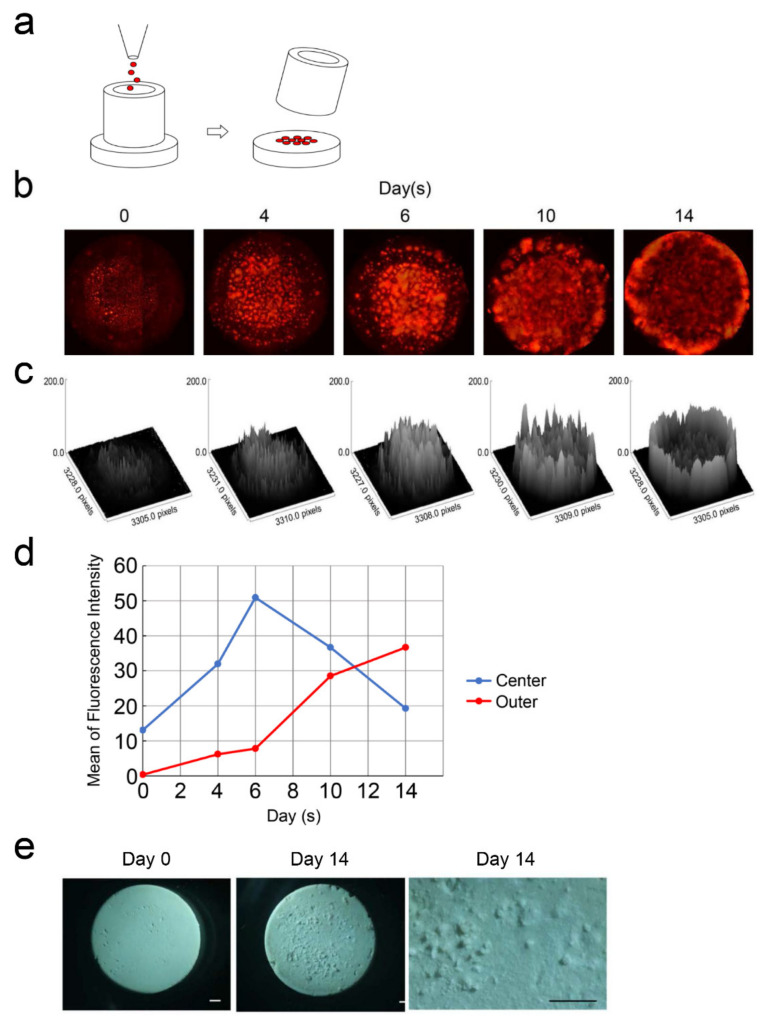
Chemotaxis of DsRed-PPU7 cells and mineralized precipitates on MTA disks. (**a**) Schema of DsRed-PPU7 cell seeding on MTA disks using a cloning cylinder. (**b**) Fluorescence microscope image of DsRed-PPU7 cells on MTA disks on Days 0, 4, 6, 10 and 14. (**c**) Three-dimensional images of fluorescence intensity of DsRed-PPU7 cells on MTA disk on Days 0, 4, 6, 10 and 14. (**d**) Changes in the fluorescence intensity of DsRed-PPU7 cells in the central or outer portion of MTA disks over time. (**e**) Stereomicroscope image of mineralized precipitates on MTA disks on Days (left) 0 and (middle) 14 and (right) a high-magnification image on Day 14 (scale bar: 1 mm).

**Figure 6 cells-09-02336-f006:**
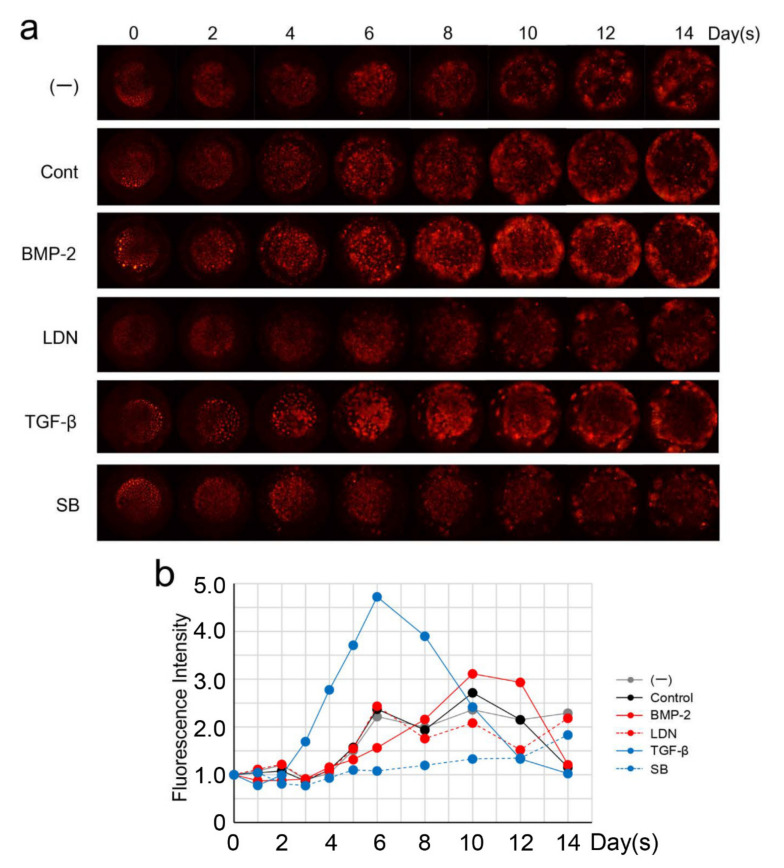
Effect of BMP-2 and TGF-β1 on the cell proliferation of DsRed-PPU7 cells on MTA disks. (**a**) Fluorescence microscope image of DsRed-PPU7 cells on MTA disks cultured in the absence (control) or presence of rhBMP-2 (BMP-2), LDN-193189 (LDN), rhTGF-β1 (TGF-β) or SB-431542 (SB). Cells were photographed every 2 days for 14 days with an exposure time of 1/20 s. DsRed-PPU7 cells were also cultured in standard medium (-) to compare with those cultured on the growth medium. (**b**) Changes over time in the fluorescence intensity of DsRed-PPU7 cells cultured under the above conditions.

**Figure 7 cells-09-02336-f007:**
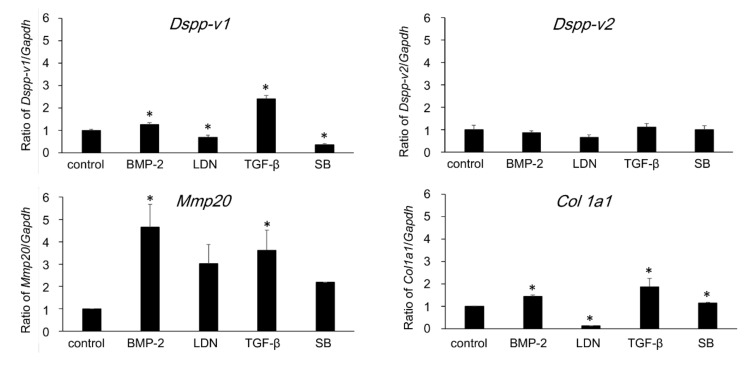
Effect of BMP-2, TGF-β1, LDN-193189 and SB-431542 on the expression of the odontoblastic marker gene in DsRed-PPU7 cells. qPCR analysis on Day 14 after culture in the absence (control) or presence of rhBMP-2 (BMP-2), LDN-193189 (LDN), rhTGF-β1 (TGF-β) or SB-431542 (SB) (*n* = 6). Dspp-v1: dentin sialophosphoprotein variant 1; Dspp-v2: dentin sialophosphoprotein variant 2; Mmp20, matrix metalloprotease 20; Col1a1: Type I collagen alpha 1 chain. All expression levels indicated an increasing or decreasing ratio when the mRNA level of the control was 1. All values are presented as the mean ± standard error (SEM); an asterisk indicates a significant difference from the control (* *p* < 0.05, Student’s *t*-test).

**Figure 8 cells-09-02336-f008:**
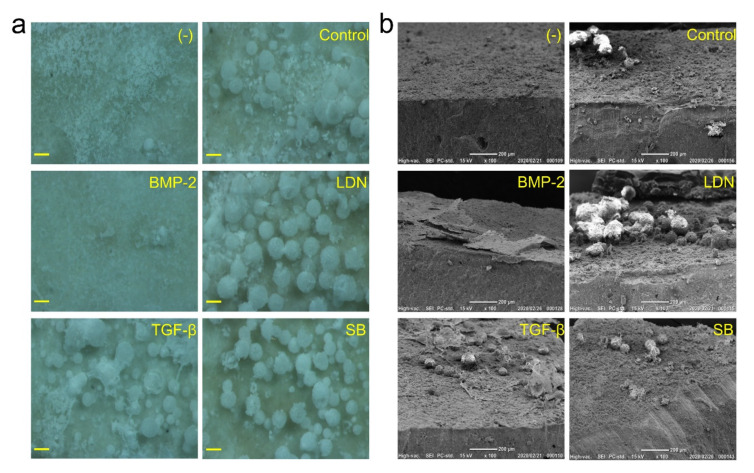
Morphological analysis of mineralized precipitates formed on MTA disks. (**a**) Stereomicroscope image and (**b**) SEM photograph of the surface of the MTA disk on Day 14 cultured with standard medium (-) or growth medium in the absence (control) or presence of rhBMP-2 (BMP-2), LDN-193189 (LDN), rhTGF-β1 (TGF-β) or SB-431542 (SB). (magnification: 100×; scale bar: 100 μm for stereomicroscope image and 200 μm for SEM).

**Figure 9 cells-09-02336-f009:**
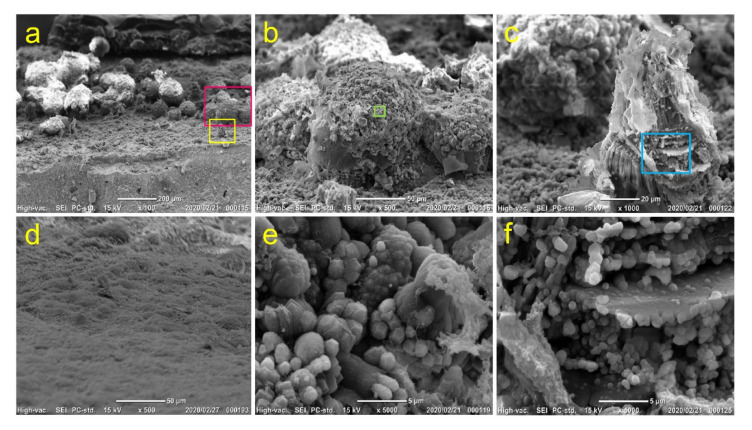
SEM observation of mineralized precipitates on MTA disks on Day 14 cultured in the presence of LDN-193189. (**a**) Surface of an MTA disk; high-magnification image of (**b**) red and (**c**) yellow boxed areas in (**a**); (**d**) MTA disk surface cultured without DsRed-PPU7 cells; high-magnification image of **(e)** green and (**f**) blue boxed areas in (**b**) and (**c**). (magnification: (**a**) 100×, (**b**,**d**) 500×, (**c**) 1000×, (**e**,**f**) 5000×; scale bar: (**a**) 200, (**b**,**d**) 50, (**c**) 20 and (**e**,**f**) 5 μm).

**Figure 10 cells-09-02336-f010:**
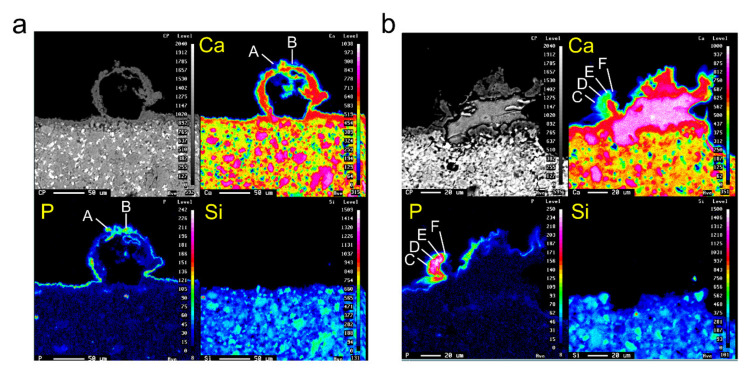
Electron-probe microanalysis (EPMA)-based elemental-mapping images of Ca, P and Si in representative mineralized precipitates. (**a**) Spherical and (**b**) triangular-pyramid-shaped precipitates (scale bar (**a**) 50 and (**b**) 20 μm). Ca/P molar ratios of each point, calculated on the basis of fluorite, were 0.82 for A, 1.15 for B, 1.95 for C, 1.63 for D, 1.69 for E and 2.01 for F.

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
