# Peer review of "Characterization of Living Dental Pulp Cells in Direct Contact with Mineral Trioxide Aggregate"

_cells, 2020, doi:10.3390/cells9102336_

Round 1

Reviewer 1 Report

The authors have tried to uncover the cellular responses of living dental pulp cells in direct contact with an MTA disk because MTA is generally used as a material for dental endodontic regenerative therapy. This is an interseting sdudy which introduces novel findings including the observation of cells on the MTA through the introduction of fluorescent protein genes. However, before publication following issues have to be addressed.

  1. Discussion section is to lengthy. I suggest concise redescription of this section.

    2. The results of this study suggest the important relationship between MTA and TGF-beta signaling for the odontobalst differentiation of pulp cells. Therefore, please discuss the relationship between MTA and TGF-beta in detail.     

Author Response

Comments and Suggestions for Authors: The authors have tried to uncover the cellular responses of living dental pulp cells in direct contact with an MTA disk because MTA is generally used as a material for dental endodontic regenerative therapy. This is an interesting study which introduces novel findings including the observation of cells on the MTA through the introduction of fluorescent protein genes. However, before publication following issues have to be addressed.

Response 1: The authors wish to thank the first reviewer for helpful and constructive comments and helpful suggestions. Our response to the reviewer’s comments is as follows:

Discussion section is too lengthy. I suggest concise redescription of this section.

Response 2: We have focused and described a discussion that is more directly related to the results of this study. Other than that, we described them in lines 469 to 498 of Appendix.

The results of this study suggest the important relationship between MTA and TGF-beta signaling for the odontobalst differentiation of pulp cells. Therefore, please discuss the relationship between MTA and TGF-beta in detail.    

Response 3: We deeply appreciate for the reviewer's very constructive comment. In this study, we found that DsRed-PPU7 cells are able to use without problems. Therefore, we have a future plan to investigate the dynamics of DsRed-PPU7 cells using dentin disk. We have changed the sentence to read, "TGF-β regulates dental pulp cell differentiation and odontoblastic differentiation, which are related to ectoderm–mesenchyme molecular interactions [47–50]. MTA contains tricalcium silicate and dicalcium silicate, and the hydration (i.e., curing reaction) of those components produces calcium hydroxide and calcium silicate hydrate. Therefore, MTA and calcium hydroxide dissolve the mineral components of dentin and it is associated with releasing TGF-β remains embedded in the matrix [51]. Subsequently, TGF-β leads to activate its signalling transduction pathways to promote the formation of tertiary dentinogenesis [52]. In some studies, calcium–silicate-containing silicon dioxide was shown to stimulate the TGF-β/Smad pathway [53,54]. The present study demonstrated that TGF-β1 increased fluorescence intensity on Day 8 in the central part of the DsRed-PPU7 cells on the MTA disk in the presence of TGF-β1. In addition to this, we demonstrated that fluorescence intensity of the central part of the MTA disk cultured in the presence of rhTGF-β1 or rhBMP-2 was almost at the same level as that of the control on Day 14. This finding may suggest that neither rhTGF-β1 nor rhBMP-2 affects the movement speed of DsRed-PPU7 cells. Since our findings indicate that TGF-β1 certainly enhances the cell proliferation, it is necessary to investigate the ability of cell proliferation in the presence of MTA using dentin disk as future studies." in lines 386 to 400 of Discussion.

Reviewer 2 Report

This is an interesting paper. The methodology is adequate (although it has some details), the research question is clinically relevant, and the paper is well written.

The result section is written in such a way that the logic of the researchers in conducting the assays is easy to understand. I liked that the authors described all the results of the experiments, including those that were not as expected but that served to plan new improved experiments. Unfortunately, this is not common in the scientific literature, although it serves to expand the knowledge on how to perform experiments avoiding waste of time and resources to other scientists.

A great number of experiments were conducted, and the images are of high-quality.

Please clarify in the methodology (not (only) in the results section) what is SB-431542 and LDN-193189 and why did you add to the medium.

Were the MTA disks sterilized? How?

A group of cells seeded on dentin disks is lacking in the experiments. This way you can say whether MTA is affecting the proliferation and differentiation of the cells or not.

The conclusion section is more a discussion than a conclusion. Please, re-write the conclusion summarizing the finding of your investigation.

Author Response

Comments and Suggestions for Authors: This is an interesting paper. The methodology is adequate (although it has some details), the research question is clinically relevant, and the paper is well written.

The result section is written in such a way that the logic of the researchers in conducting the assays is easy to understand. I liked that the authors described all the results of the experiments, including those that were not as expected but that served to plan new improved experiments. Unfortunately, this is not common in the scientific literature, although it serves to expand the knowledge on how to perform experiments avoiding waste of time and resources to other scientists.

A great number of experiments were conducted, and the images are of high-quality.

Response 1: The authors wish to thank the second reviewer for helpful and constructive comments and helpful suggestions. Our response to the reviewer’s comments is as follows:

Please clarify in the methodology (not (only) in the results section) what is SB-431542 and LDN-193189 and why did you add to the medium.

Response 2: We have changed the sentence to read, "50 nM LDN-193189 (Tocris Bioscience, Bristol, UK) which is a selective BMP signaling inhibitor, 1 ng/mL of human recombinant transforming growth factor β (rhTGF-β), or 1 μM SB-431542 (ChemScene, Monmouth Junction, NJ, USA) which is a selective inhibitor of TGF-β Type I receptor activin receptor-like kinase A" in lines 125 to 128 of Materials and Methods.

The reason why we added LDN-193189 or SB-431542 in the medium was to inhibit the activity of endogenous BMP-2 and TGF-β that might be contained in cells and/or medium.

Were the MTA disks sterilized? How?

Response 3: Yes, they were sterilized. We have added the sentence to read, "Prior to the preparation of MTA disk, glass paste plates, spatulas, and silicone molds for kneading were immersed in 100% ethanol and ultrasonically treated for 5 min in a fume food." in lines 94 to 96 of Materials and Methods.

A group of cells seeded on dentin disks is lacking in the experiments. This way you can say whether MTA is affecting the proliferation and differentiation of the cells or not.

Response 4: We deeply appreciate for the reviewer's very constructive comment. In this study, we found that DsRed-PPU7 cells are able to use without problems. Therefore, we have a future plan to investigate the dynamics of DsRed-PPU7 cells using dentin disk. We have changed the sentence to read, "TGF-β regulates dental pulp cell differentiation and odontoblastic differentiation, which are related to ectoderm–mesenchyme molecular interactions [47–50]. MTA contains tricalcium silicate and dicalcium silicate, and the hydration (i.e., curing reaction) of those components produces calcium hydroxide and calcium silicate hydrate. Therefore, MTA and calcium hydroxide dissolve the mineral components of dentin and it is associated with releasing TGF-β remains embedded in the matrix [51]. Subsequently, TGF-β leads to activate its signalling transduction pathways to promote the formation of tertiary dentinogenesis [52]. In some studies, calcium–silicate-containing silicon dioxide was shown to stimulate the TGF-β/Smad pathway [53,54]. The present study demonstrated that TGF-β1 increased fluorescence intensity on Day 8 in the central part of the DsRed-PPU7 cells on the MTA disk in the presence of TGF-β1. In addition to this, we demonstrated that fluorescence intensity of the central part of the MTA disk cultured in the presence of rhTGF-β1 or rhBMP-2 was almost at the same level as that of the control on Day 14. This finding may suggest that neither rhTGF-β1 nor rhBMP-2 affects the movement speed of DsRed-PPU7 cells. Since our findings indicate that TGF-β1 certainly enhances the cell proliferation, it is necessary to investigate the ability of cell proliferation in the presence of MTA using dentin disk as future studies." in lines 386 to 400 of Discussion.

The conclusion section is more a discussion than a conclusion. Please, re-write the conclusion summarizing the finding of your investigation.

Response 5: We have tried to re-write the conclusion summarizing our finding in lines 435 to 443 of Conclusions. In addition, we have moved a part of conclusion to read, “In research on MTA, approximately 60 papers have been reviewed by a comprehensive literature survey via electronic database of PubMed/MEDLINE [32]. In studies using MTA disk, the effects of cells on elution components from disks in the same environment as that of the cells and disks with a culture insert were investigated [19,23,33]. In addition, research was carried out to examine the effects of cells seeded on the plate on eluants extracted by immersing the disks in a culture medium [21]. However, there were few reports that directly show the dynamics of living cells in contact with MTA because the direct observation of cells is difficult, as the MTA is an impermeable cured body. Our challenge was to directly observe living PPU7 cells on an opaque MTA disk.” in lines 338 to 346 of Discussion.

Round 2

Reviewer 2 Report

The recommendations/comments were addressed properly